**DOI: 10.1038/ncomms13735**　**OPEN**

# Insect mimicry of plants dates back to the Permian

Romain Garrouste[1,*], Sylvain Hugel[2,*], Lauriane Jacquelin[1], Pierre Rostan[3], J.-Sébastien Steyer[4], Laure Desutter-Grandcolas[1,**] & André Nel[1,**]

In response to predation pressure, some insects have developed spectacular plant mimicry strategies (homomorphy), involving important changes in their morphology. The fossil record of plant mimicry provides clues to the importance of predation pressure in the deep past. Surprisingly, to date, the oldest confirmed records of insect leaf mimicry are Mesozoic. Here we document a crucial step in the story of adaptive responses to predation by describing a leaf-mimicking katydid from the Middle Permian. Our morphometric analysis demonstrates that leaf-mimicking wings of katydids can be morphologically characterized in a non-arbitrary manner and shows that the new genus and species *Permotettigonia gallica* developed a mimicking pattern of forewings very similar to those of the modern leaf-like katydids. Our finding suggests that predation pressure was already high enough during the Permian to favour investment in leaf mimicry.

[1] Institut de Systématique, Évolution, Biodiversité, ISYEB, UMR 7205, CNRS, MNHN, UPMC, EPHE, Muséum national d'Histoire naturelle, Sorbonne Universités, 57 rue Cuvier, CP 50, Entomologie, F-75005, Paris, France. [2] INCI, UPR 3212 CNRS, Université de Strasbourg, 8 rue Blaise Pascal, 67084 Strasbourg, France. [3] Mines and Avenir, Les Albrands, F-05380 Châteauroux Les Alpes, France. [4] Centre de Recherches en Paléobiodiversité et Paléoenvironnements, UMR 7202—CNRS, MNHN, UPMC, EPHE, Muséum national d'Histoire naturelle, Sorbonne Universités, 8 rue Buffon, CP 38, F-75005 Paris, France. * These authors contributed equally to this work. ** These authors jointly supervised this work. Correspondence and requests for materials should be addressed to R.G. (email: garroust@mnhn.fr) or to A.N. (email: anel@mnhn.fr).

Plant mimicry occurs in many clades of insects, with the most striking cases in butterflies, praying mantises, stick insects and katydids. In these groups, mimicry involves spectacular modifications of the body and wings, in response to predation. However, the accurate fossil record of plant mimicry is limited to a few Cenozoic and Mesozoic taxa[1–4], and is sometimes difficult to interpret[5,6]. In contrast, disruptive patterns of wing coloration (non-homomorphous alternation of hyaline and dark transverse bands on wings) are frequent among Upper Carboniferous flying herbivorous insects (viz. Palaeodictyoptera[7]), possibly because at this time the main predators of flying insects were only flying insects[8]. Cases of disruptive coloration are comparatively rare among Carboniferous and Early Permian insects living in the vegetation (viz. Dictyoptera, Orthoptera)[9]. Gliding and terrestrial vertebrates (for example, amphibians, synapsids and sauropsids) appeared or became very diverse in the Middle Permian[8,10], in addition to the arthropod predators that were already present during the Carboniferous, but apparently without any particular impact on the diversification of mimicry strategies among the insect prey. This phenomenon was supposed to happen during the Triassic and the Jurassic[9].

The present discovery of a Guadalupian forewing, representing the oldest katydid fossil, contradicts this view. It demonstrates that the leaf-like homomorphous cryptic mimicry was already present >100 million years (Ma) before the previous oldest records. This wing displays the same modifications in shape and venation as seen in modern leaf-like katydids. It also attests that Tettigonioidea are much older than previously thought, as this clade was considered as not older than the Jurassic[11–13].

## Results

### Systematic palaeontology.

<div align="center">

Insecta Linné, 1758

Orthoptera Olivier, 1789

Ensifera Chopard, 1920

Tettigonioidea Krauss, 1902

Permotettigoniidae, Nel & Garrouste fam. nov.

*Permotettigonia gallica*, Nel & Garrouste gen. et sp. nov.

</div>

**Etymology.** The genus name refers to Permian and *Tettigonia*; the species name refers to the Latin name for France.

**Holotype.** MNHN-LP-R 63853, Muséum National d'Histoire Naturelle, Paris, France.

**Horizon and locality.** Red siltsones, Cians Formation, Roadian, Middle Permian; Dôme de Barrot, 760 m alt., Roua Valley (Supplementary Fig. 1, Supplementary Note 1), village of Daluis, Alpes Maritimes, France.

**Diagnosis.** The fossil is a nearly complete and very broad tegmen (forewing) (Fig. 1) with a very broad and strongly corrugate field between vein subcostal posterior (ScP) and anterior wing margin, crossed by numerous long veinlets alternatively convex and concave, all nearly perpendicular to ScP and to anterior wing margin. A second broad field is present between radial vein (R) and median vein (M) and a third between M and cubital complex vein (CuA + CuPaα); all crossed by long parallel veinlets more or less curved and perpendicular to main longitudinal veins, but not alternatively convex and concave as in subcostal field, crossveins present but poorly preserved in field between R and M; M and CuA + CuPaα simple and distally fused again; cubitus posterior (CuP) divided into branches CuPaα, CuPaβ and CuPb, unlike in modern katydids (Supplementary Fig. 2); anal vein A1 straight (extended description in Supplementary Note 2). *Permotettigonia* is the oldest record

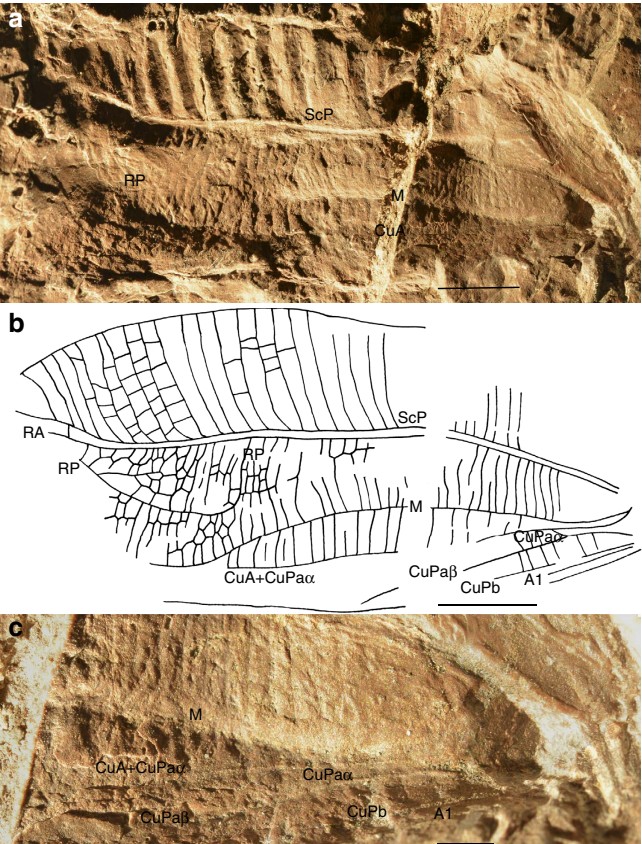

**Figure 1 | *Permotettigonia gallica* gen. et sp. nov.** (**a**) General habitus of tegmen imprint. (**b**) Reconstruction of tegmen. (**c**) Detail of cubito-anal field. Scale bar, 5 mm (**a**,**b**), 2 mm (**c**).

of the katydid clade (for discussion on the other candidates see Supplementary Note 3).

## Discussion

The forewing of *Permotettigonia* is very broad compared with its length, only *ca.* 2.7 times longer than wide, with its surface separated into two main fields of nearly the same width (subcostal field and field between R and CuA of the same width, both 6.0 mm wide), separated by a pair of parallel longitudinal strongly marked veins ScP and R, secondary veinlets in subcostal and median fields nearly perpendicular to main veins, with those of subcostal field emerging obliquely from ScP and making a strong bend just after their base, and subcostal field corrugated. The presence of a pronounced angle between the anal field and the rest of the tegmen suggests that the tegmina of *Permotettigonia* were not held horizontally but perpendicular to the body. All these structures mirror those of some modern leaf-like macropterous Tettigoniidae[14] that have an appearance of upright leaves (case of occultation of the volume as relief, surface and outline), for example, *Pterophylla*[15,16], suggesting a similar situation for *Permotettigonia*.

To examine the possibility that the tegmina of *Permotettigonia* could display a leaf-mimicking camouflage, we compared it with morphometric descriptors distinctive of the modern leaf-mimicking katydids. As such morphometric descriptors were not already available; we performed a comprehensive morphological analysis of the tegmina of modern katydids (data set in Supplementary Table 1). Pterochrozinae is the only tettigoniid

subfamily including only leaf-mimicking species[17,18], which are considered the most perfectly camouflaged katydids. This subfamily was therefore used to epitomize leaf-mimicking species.

We carried out morphometric analyses in a set of representative modern tettigoniids, including Pterochrozinae and other subfamilies with univariate (non-linear regression and curve-fitting), bivariate (correlation between variables) and multivariate (Principal Component Analysis or PCA) approaches, based on size-independent measures (ratios) to focus on shape analysis.

These analyses allowed to distinguish significantly two wing morphologies (Fig. 2; Supplementary Figs 3–8): one is corresponding to the tegmina of genera known for displaying leaf mimicry (including Pterochrozinae), and the second to the non-mimetic tegmina. Distributions of morphological descriptors were significantly better fitted with two components models having one component fixed to values obtained from Pterochrozinae, than with one-component models (Supplementary Table 2). For all parameters, treated with non-linear regression models and curve-fitting, that allow modern leaf-mimicking species to be distinguished from other katydids, *Permotettigonia* displayed values shared by at most 3% of the modern non-leaf-mimicking species and by up to 30% of modern leaf-mimicking species (Fig. 2a,b, Supplementary Table 3, Supplementary Figs 3–

6).As the radial vein is strongly zigzagged and not thickened in *Archepseudophylla*, parameters involving this vein are not relevant to analyse this taxon. Both the circularity and the ratio width/length, which do not involve the radial vein, confirm that this fossil is close to the morphology of extant mimetic species, but relatively less than the Permian fossil. In the morphospace of tettigoniid tegmina using PCA, it is very close to *Permotettigonia* inside the mimetic group as defined by Mugleston *et al.*[14] (Fig. 2c).

Using PCA that allows visualizing the set of all variables in the same space (or morphospace analysis), *Permotettigonia* is close to the centre of the set of leaf-mimicking species as defined by Mugleston *et al.*[14] (Fig. 2c). Furthermore, if we consider a PCA based on the different subfamilies, *Permotettigonia* is between the Pterochrozinae and the Pseudophyllinae; two clades that mainly comprise leaf-mimicking species (see Supplementary Fig. 7).

The morphometric results strongly support the leaf mimicry function of the tegmina of *Permotettigonia*. The difficulty is to determine which kind of leaves *Permotettigonia* could have mimicked. Of course the angiosperms were not present during the Permian, but plants with very similar leaf or pseudoleaf morphology were present, symmetrical to a midvein, and with subperpendicular second order veins and marked corrugations. It is especially the case of the very large angiosperm-like leaves of the Gigantopterideae and of some other tracheophytes lineages[19–21].

Unfortunately, no plant is recorded from the Cians Formation. However, two red Middle Permian palaeofloras of similar ages are known, from the Bau Rouge Member (Kungarian–Roadian[22,23], Toulon Basin, Var), and the Mérifons Member (Kungurian–Capitanian[23,24], Salagou Formation, Lodève Basin, Hérault), at ∼150 and 350 km from the Dome de Barrot. All these outcrops are corresponding to shallow playa lakes and submerged flood plains[25].

In the Bau Rouge Member and the Salagou Formation, the *Taeniopteris* leaves (a type of plant ranging between the Late Paleozoic to the Mesozoic) are one of the best-known candidates for a mimicry by *Permotettigonia* because they can be up to 5 cm wide with the surface presenting a series of corrugations perpendicular to a strong midvein (Supplementary Fig. 10), quite similar to the corrugations of the wing of *Permotettigonia*[26]. Also, the second-order veins of the leaves of *Taeniopteris* are separating obliquely from the main midvein and becoming more or less perpendicular to the margin, quite similarly to the veinlets in the subcostal field of *Permotettigonia*. The other possible problem is the length of the leaf (up to 22 cm long). Nonetheless, some modern katydids can be mimetic with leaves that are substantially longer than the insect, for example, *Segestidea* living on *Calamus* palms[27]. Moreover, the leaves of *Taeniopteris* were also attacked by insects in the Permian, having margin feedings attributed to orthopteroid insects[20,28–30].

*Permotettigonia* may have been both an imitator of living plants as well as parts of leaves among which it could go unnoticed, especially if it had also a cryptic coloration (homochromy), a character that unfortunately remains unknown (see possible reconstruction in Fig. 3). It is likely that this mimicry conferred an advantage in avoiding detection by predators, such as flying insects (like giant griffenflies, also recorded from the Dôme de Barrot[31]) and terrestrial tetrapods.

During the Late Carboniferous and the Early Permian, the pressure of predation on the palaeopteran flying insects by land vertebrates was probably relatively low compared with that by carnivorous flying insects, such as giant griffenflies[31]. The plant–insect homomorphy is not recorded at that time and disruptive coloration was relatively rare among the neopteran insects that were living among plants (Dictyoptera, Orthoptera)[9].

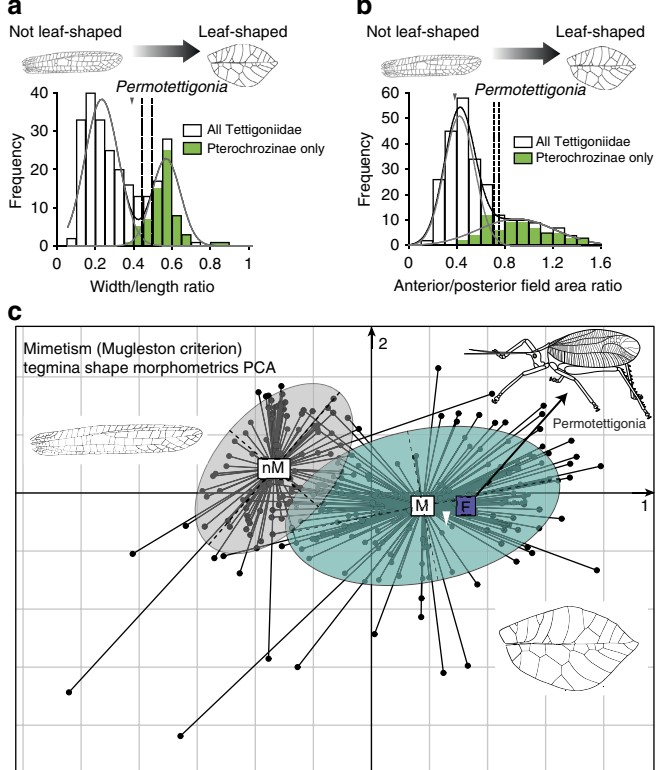

**Figure 2 | Morphometric analyses of leaf-mimicking versus non-leaf-mimicking tegmina.** (**a,b**) Non-linear regression modeling. (**a**) Tegmina width/length ratio. (**b**) Anterior field area/posterior field area ratio. (**c**) Morphospace of tettigoniid tegmina using PCA, Ordination of 253 modern Tettigoniidae, the Cenozoic *Archepseudophylla* fossils Nel *et al.*, 2008, plus *Permotettigonia gallica*, based on five morphometric indices defined in Supplementary Information. Taxa plotted in a factorial map divided into two confidence ellipses using Mugleston *et al.*[14] criterion, nM centroid of cloud of non-mimetic taxa, M centroid of cloud of mimetic taxa. F *Permotettigonia gallica*. Values for *Permotettigonia* are indicated by grey triangles, and white triangle for *Archepseudophylla*.

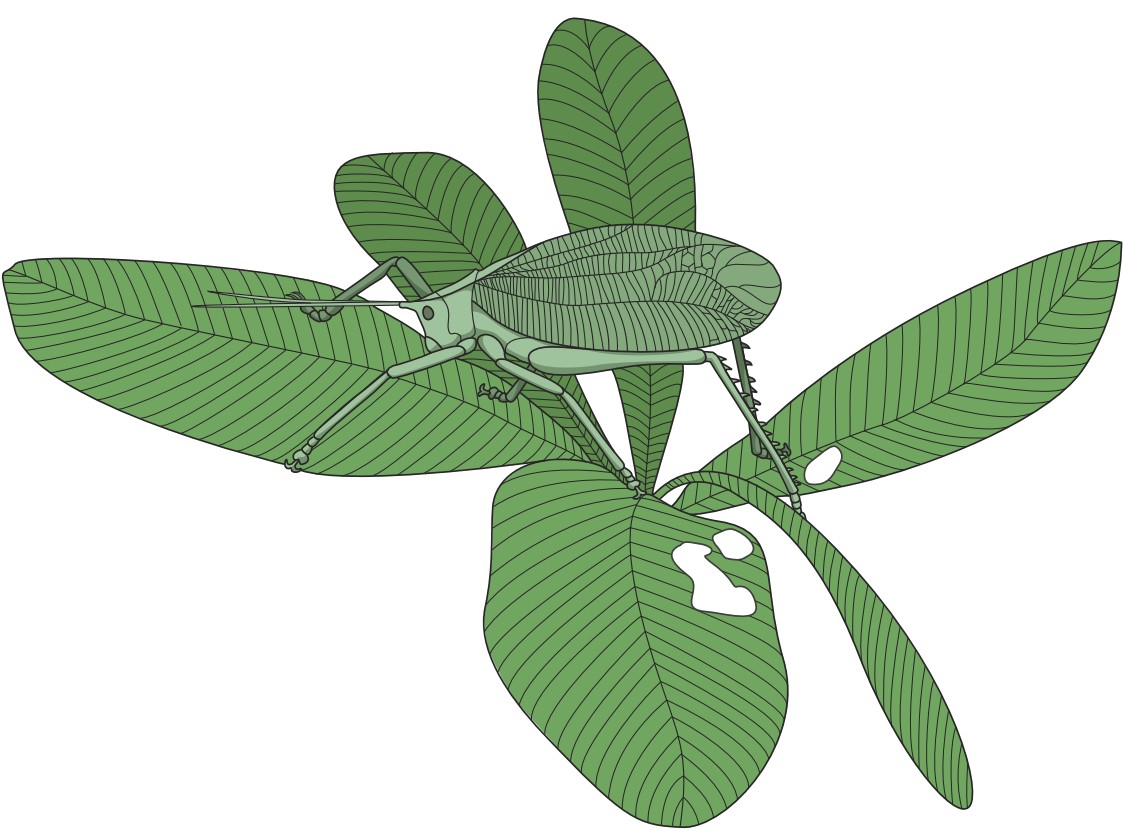

**Figure 3 | Reconstruction of *Permotettigonia gallica* gen. et sp. nov. on *Taeniopteris* sp.** Body interpreted after a *Pterophylla* sp. with the tegmen of *P. gallica* (copyright C. Garrouste).

The Middle Permian diversification of the small insectivorous gliding and terrestrial vertebrates probably exercised a strong selective pressure on their potential prey[8], sufficient to drive the first cases of acquisition of homomorphy with plants. Disruptive coloration became also frequent among the polyneopteran insects living among plants during the Triassic[9]. The fact that the fossil record of disruptive coloration is clearly older than homomorphy with plants may suggest that the former strategy has lower metabolic costs and/or is much easier to evolve. The two strategies (disruptive coloration versus homomorphy) were maintained in many clades (Orthoptera, Odonata and so on) from the Mesozoic to the present. Many modern insects combine these strategies, also in association with specialized behaviors, to reduce the risk of predation[32,33].

## Methods

**Preparation, observation and description.** The specimen was prepared using sharp spines. Photographs were taken with a Nikon D800 digital camera with AF-MicroNikkor 60 mm 2.8, with dry specimens and line drawings were prepared using a camera lucida on an Olympus SZX-9 stereomicroscope. Original photographs were processed using the image-editing software Adobe Photoshop CS6. Standard wing venation nomenclature is followed for Orthoptera[9]. The diagnoses given in the main text are the same for the familial, generic and specific levels because all are presently monospecific.

**Morphometric analyses.** Our approach is based on a data set based on 253 tettigoniid species that are representative of morphological disparity of macropterous forms. We also added the Cenozoic leaf-mimetic genus *Archepseudophylla*. Samples of all modern tettigoniid subfamilies were included in the analysis, except exclusively micropterous or apterous subfamilies. The established morphospace expresses a gradient from forms considered non-mimetic towards forms that are extremely mimetic of leaves of angiosperms (morphological patterns, colours and associated behaviour). Measurements were performed with ImageJ 1.46r (NIH) software, on drawings or pictures of tegmina, either from collection specimens (S.H. collection, or Musée Zoologique de Strasbourg)

or available on OSF site[32]. As all parameters used for the analysis were dimension-free ratios calculated after our measurements, no scaling of the used images was required. The measurements taken to calculate the ratios corresponded to general shape descriptors, with no hypothesis on homologies, except when defining the anterior and the posterior fields of the tegmen, where the radius was considered as the limit between these fields.

The measurements taken on the tegmen were: maximal length; width on the middle; anterior field width; posterior field width; total area; anterior field area; posterior field area; total perimeter; anterior field perimeter; posterior field perimeter. With these measurements, the following four dimension-free parameters were calculated: (width/length) ratio; (anterior field width/posterior field width) ratio; (anterior field area/posterior field area) ratio; and [$4\pi$(anterior field area)/(anterior field perimeter)$^2$] corresponding to anterior field circularity. Tegmina were also coded as a single binary character, as being either leaf-like or not leaf-like following the criterion of Mugleston et al.[14], that is, 'leaf-like tegmina were defined as being oblong with the maximum width of the wing larger than the height of the thorax, or not leaf-like, with narrow forewings that are not wider than the height of the thorax'. The values of the five parameters are listed in Supplementary Table 1.

The situation in our fossil is particular because the tip of its tegmen is missing. Two kinds of measurements were taken: measurements with no extrapolation and measurements with missing parts extrapolated with circle segment connections (Supplementary Fig. 9). The parameters calculated with measurements of *P. gallica* were then compared with the distribution of these parameters in each type of tegmina (leaf-mimicking and non-leaf mimicking). The values for the Mugleston et al.[14] criterion are unknown for *P. gallica* as its thorax is not preserved.

**Nonlinear regression analyses.** Analyses were performed with KyPlot 2.15 (KyensLab). Nonlinear regression allows defining if data distributions are significantly best-described with one or with more components. We used this approach to assess whether morphologically distinct types of forewings can be distinguished and whether one of the forewing types includes all Pterochrozinae, a subfamily epitomizing leaf-mimicking species.

The cumulative distribution of each dimension-free parameter calculated with measurements described above was fitted using nonlinear regression[34]. Cumulative distribution was used to avoid the loss of information induced by data binning.

The cumulative distribution of each parameter of all measured Tettigoniidae was first fitted with the following single-cumulative distribution function of the normal distribution.

One component model:

$$F(x, A1, A2) = \frac{1}{2}\left(1 + \mathrm{erf}\left(\frac{x - A1}{A2 \times \sqrt{2}}\right)\right) \tag{1}$$

where $A1$ is the mean; $A2$ is the standard deviation; erf is Gauss error function.

To define characteristics of epitomized leaf-mimicking forewings, we fitted the cumulative distribution of each parameter from Pterochrozinae with the same cumulative distribution function of the normal distribution (equation (1)). We considered the parameters obtained with that fitting as characteristic of leaf-mimicking tegmina.

The mean ($A1$) obtained from that single-component fitting of Pterochrozinae parameters was then forced as the mean one of the two components of the following function used to fit each parameter of all measured Tettigoniidae.

Two components model, one component forced to Pterochrozinae mean:

$$F(x, A1, A2, A3, A4, A5) = \frac{1}{2}\left[A5 \times \left(1 + \mathrm{erf}\left(\frac{x - A3}{A4 \times \sqrt{2}}\right)\right)\right. \\ \left. + (1 - A5) \times \left(1 + \mathrm{erf}\left(\frac{x - A1}{A2 \times \sqrt{2}}\right)\right)\right] \tag{2}$$

where $A1$ is the mean obtained by fitting Pterochrozinae parameters with equation (1); $A2$ is the standard deviation of the component with forced $A1$; $A3$ and $A4$ are, respectively, the mean and the s.d. of the other component; $(1 - A5)$ is the relative contribution of the component forced to Pterochrozinae mean; $A5$ is the contribution of the other component.

We then compared which of the single component model or the two components model (equation (1) versus equation (2)) was best describing the distributions of all measured Tettigoniidae using the extra sum-of-squares $F$ test with a threshold of significance set at $P < 0.01$, as well as decreases in Akaike's Information Criterion and Bayesian Information Criterion. Should the two components model be significantly better than the single-component model, it would allow defining the mean and the s.d. of each parameter for the two types of tegmina: leaf-mimicking and non-leaf-mimicking.

Although cumulative distribution of the parameters was used for the analysis, binned data were represented in the figures together with a Gaussian distribution having the parameters obtained with the fitting of the cumulative distribution.

**Morphospace representation using multivariate statistic ordination.** A morphospace is a practical or theoretical mathematical space for study the phenotypes of living or fossil organisms, widely used in zoology, botany and evolutionary biology[35–37]. In theoretical morphospaces, the axes of the reduced space are determined by a small set of parameters of morphogenetic or other biological models, derived from theoretical considerations rather than from the organisms themselves. In morphometric (or empiric) morphospaces, it is often not trivial to choice a particular type of variables and metrics that properly reflect the studied developmental, functional, or evolutionary properties and traits of the organisms. This choice is a compromise between accessible characters and scope. For fossil studies, this is particularly delicate as it is dependent of the quality of preservation. The structure of a morphospace is also determined by the choice of the characters to include, by the way these characters are coded, and by the method of visualization/construction of the space[36]. The PCA ordination method[38] is a widely used multivariate statistics method which creates a low-dimension ordination that maximizes the variation between measured specimens[17,19], and creates a visualization by a projection (two or three axes, or 3D outputs). The PCA was performed using R software, package ADE4 (refs 39–41) and RStudio[42]. Graphic displays have been modified in Adobe Illustrator CS6.

The PCA was performed using R software, package ADE4 (ref. 41).

**Nomenclatural acts.** This published work and the nomenclatural acts it contains have been registered in ZooBank, the proposed online registration system for the International Code of Zoological Nomenclature (ICZN). The ZooBank LSIDs (Life Science Identifiers) can be resolved and the associated information viewed through any standard web browser by appending the LSID to the prefix http://zoobank.org/. The ZooBank LSIDs (Life Science Identifiers) for the new family, genus and species are as follows: urn:lsid:zoobank.org:act:831FB4B9-A27A-4A43-989E-D74F26B42C19, urn:lsid:zoobank.org:act:654E7DAC-083B-4F43-B32B-8E4DE28F3877 and urn:lsid:zoobank.org:act:5B6A3647-A015-46D2-BC60-E0B826C2BDE2.

**Data availability.** All relevant data are available from the authors. The specimen MNHN-LP-R 63853 is housed at the Muséum National d'Histoire Naturelle (MNHN), Paris, France.

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

## Acknowledgements

This work was supported by a grant from Agence Nationale de la Recherche under the LabEx ANR-10-LABX-0003-BCDiv, in the program 'Investissements d'avenir' ANR-11-IDEX-0004-02 (Labex BCDiv 'Diversité Biologique et Culturelle') and by the ATM 'Blanc' MNHN 2015-2016. Many thanks to C. Garrouste for the reconstruction of *Permotettigonia gallica* and to S. Fouché (Musée de Lodève) for the photograph of the plant *Taeniopteris* sp. from Lodève. Many thanks to B. Warren (University of Zurich) for judicious and important comments on the preliminary version of the paper. We also thank the 'Musée Zoologique de la Ville et de l'Université de Strasbourg' for giving access to their collections and J.L. Rodeau for discussions on non-linear regression modelling.

## Author contributions

R.G. and S.H. performed the morphometric analyses; A.N., L.D.-G., S.H., R.G., P.R., J.-S.S., L.J. prepared the manuscript; S.H., L.J. and R.G. prepared the figures, P.R. discovered the fossil; A.N. and R.G. designed the program; L.D.-G. and A.N. jointly supervised this work.

## Additional information

**Publisher's note**: 

