## [Peer Review File · Nature Communications]

Reviewers' comments:

Reviewer #1 (Remarks to the Author):

The manuscript is well written, and deserves publication in Nature Communications. It represents an important contribution for understanding early insect-plant associations and develops the insect-plant mimicry more than 100 million years than the present knowledge. However, the manuscript needs detailed corrections before its final publication.

1. The line drawing of tegmen (Fig. 1b; I do think this drawing is a "reconstruction"): from my view of the photos, the tegmen seems to preserve more parts than what the line drawing shows. In basal Subcosta region it appears to have a rather complete wing base with a few non-specialized veins. In median Radius region, it is clearly displayed the cross veins specialized to a serial of successive and oblique veins as "reflection" of those in Sc region. This character missed in the drawing in RP region and unsuccessful before RP (please check again).
2. The reconstruction (Fig. 3): the comb-like, oblique and successive cross-veins in Radius region seem to rather optionally drawn, unlike what the fossil really displays as a leaf shape, like some ferns such as *Glossopteris* or *Gigantopteris*. I would suggest the authors weaken some veins, such as RP, M and Cu veins in reconstruction, and clearly mark RA, which is an interesting character that makes the central vein much thicker.
3. The manuscript presents an excellent work for orthopteran interpretations based on both fossil and Recent forms. However, the section concerning Permian plants needs slightly to be improved. I understand that this insect locality has not yielded lots of plant fossils. The adjacent localities produced some small leaves (S. Fig 10) indeed, presenting several critical characters for such mimicry relationship, but not the size and shape. There are lots of Permian large fern leaves displaying a morphology of an elongate elliptic shape with a central vein and lateral comb-like secondary veins.
4. Reference 2 and 3: Reference 3 should be Wang et al., the MS lost the rest co-authors; the sequence is incorrect, 2 and 3 should be switched each other; Yong-jie, not Yon-gjie. I would suggest to remove the reference 2. I do not think it is true mimicry. Those mecopterans from Daohugou displayed the normal pose with four wings slightly stretched out, as many mecopteran fossils in their preservation. It partly resembles ginkgo leaves but the major rejection due to their venation without any specialization. Moreover, those Daohugou mecopterans exhibit a constriction at wing base, an unusual character to be distinguished from ginkgo leaves.
5. Linguistic and format mistakes and other suggestions: 1) The authors may use more than 100 Ma instead of ca. 100 Ma, due to the age of Daohugou (with the previous earliest insect-plant mimicry fossil) no more than 165 Ma, and 265 Ma is just the boundary of Capitanian and Wordian; 2) demonstrates that the leaf-like homomorphous defense strategy was already... not necessarily "defense strategy", and it should be regarded as crypsis, or cryptic mimicry; 3) the Middle Permian should be change to Guadalupian. 4) SUPPLEMENTARY INFORMATION: The fossil was discovered by one of us (R.P.), should be P. R.; 5) in the red Permian of the Cians Formation... should be reworded; 6) in table 1, the ratio of length/width etc. is "too accurate" that present 15 numbers after radix point.

Reviewer #2 (Remarks to the Author):

The present manuscript deals with fascinating discovery of the earliest evidence of leaf-mimicking katydid from the Middle Permian deposits in France. This peculiar group of orthopteran insects was so far unknown from the Paleozoic strata and their oldest members come from Mesozoic. In addition the supportive morphometric analyses of fore wings (tegminae) based on extensive dataset of extant representatives of tettigoniids are given. PCA visualization of recent leaf-mimicking katydid subfamilies enabled Permotettigonia placing in the intersection of Pseudophyllinae and Pterophyllinae. Thus, the present contribution is original and novel in term of the earliest record of plant insect mimicry and parallel application of such complex landmark analysis of tegminae of recent katydid mimicking species.

1/ I recommend extending your review by indication of the taxa with supposed crypsis in other Paleozoic groups like particularly roachoids (Dictyoptera) with pteridosperm pinnules, other considered orthopterans (see Jarzembowski 1994, Prokop et al. 2015) and Triassic hemipterans (Cicadomorpha) presumably imitated buds, bracts, thorns, seeds, seed-bearing organs, or other small parts of their host plants (Shcherbakov 2011). Some of these putative members were put in doubt, but it should be briefly noticed in this contribution. Authors should note in their summary the earlier work dealing with an exceptional Cenozoic genus *Archepseudophylla* of leaf-mimicking katydid known from France (Nel et al. 2008). Perhaps it could be interesting to incorporate this well preserved Cenozoic genus in your landmark analysis and visualizes by PCA to see the position of another fossil genus.

2/ I would prefer to cite several concrete examples with disruptive pattern of wing coloration like *Sinodunbaria* (Palaeodictyoptera) (e.g., Li et al. 2013) etc. instead of reference to the general textbook where colour patterns are omitted.

3/ I suggest keeping diagnosis in article as set of characters only and transfer the remaining comparison with other taxa into discussion as usual. Thus the main text will be clearly arranged.

To sum up, this work is an important contribution to our knowledge of plant insect ecology sparsely documented in Paleozoic. Besides a few above mentioned suggestions the manuscript is excellent. I fully support acceptance as publication in Nature Communication journal.

Please consider citing of the following papers:

Jarzembowski EA. 1994. Fossil cockroaches or pinnules insects? Proceedings of the Geologists' Association. 105:305–311.

Li Y., Ren D., Pecharová M., & Prokop J. 2013. A new palaeodictyopterid (Insecta: Palaeodictyoptera: Spilapteridae) from the Upper Carboniferous of China supports a close relationship between insect faunas of Quilianshian (northern China) and Laurussia. *Alcheringa* 37(4):487-495.

Nel, A., Prokop, J. & Ross, A.J. 2008. New genus of leaf-mimicking katydids (Orthoptera: Tettigoniidae) from the Late Eocene - Early Oligocene of France and England. *C.R. Palevol*, 7 (4): 211-216.

Prokop J., Szwed J., Lapeyrie J., Garrouste R. & Nel A 2015. New middle Permian insects from Salagou Formation of the Lodève Basin in southern France (Insecta: Pterygota). *Annales de la Société Entomologique de France* 51(1): 14-51.

Shcherbakov DE. 2011. New and little-known families of Hemiptera Cicadomorpha from the Triassic of Central Asia - early analogs of treehoppers and planthoppers. *Zootaxa* 2836:1–26.

Reviewers' comments:

Reviewer #1 (Remarks to the Author):

The manuscript is well written, and deserves publication in Nature Communications. It represents an important contribution for understanding early insect-plant associations and develops the insect-plant mimicry more than 100 million years than the present knowledge. However, the manuscript needs detailed corrections before its final publication.

1. The line drawing of tegmen (Fig. 1b; I do think this drawing is a "reconstruction"): from my view of the photos, the tegmen seems to preserve more parts than what the line drawing shows. In basal Subcosta region it appears to have a rather complete wing base with a few non-specialized veins.

We thank the reviewer for his careful examination of the fossil. We agree that the basal part of the subcostal area may well be present; we therefore indicated it in the supplementary information. We nevertheless preferred to limit our reconstruction to doubtless elements

In median Radius region, it is clearly displayed the cross veins specialized to a series of successive and oblique veins as "reflection" of those in Sc region. This character missed in the drawing in RP region and unsuccessful before RP (please check again).

Here again, the reviewer may be right, but as for the subcostal region, we do not feel confident to draw it. But we agree that this has to be mentioned, and we added it in the description about the shape of the veinlets between R and M

2. The reconstruction (Fig. 3): the comb-like, oblique and successive cross-veins in Radius region seem to rather optionally drawn, unlike what the fossil really displays as a leaf shape, like some ferns such as Glossopteris or Gigantopteris. I would suggest the authors weaken some veins, such as RP, M and Cu veins in reconstruction, and clearly mark RA, which is an interesting character that makes the central vein much thicker.

As we are not absolutely sure that these crossveins are really present, we prefer to avoid to indicate them, RA is not really stronger than ScP, so we prefer to keep it as it is

3. The manuscript presents an excellent work for orthopteran interpretations based on both fossil and Recent forms. However, the section concerning Permian plants needs slightly to be improved. I understand that this insect locality has not yielded lots of plant fossils. The adjacent localities produced some small leaves (S. Fig 10) indeed, presenting several critical characters for such mimicry relationship, but not the size and shape. There are lots of Permian large fern leaves displaying a morphology of an elongate elliptic shape with a central vein and lateral comb-like secondary veins.

Unfortunately, but the absence of complete flora in the 'red Permian' of the south of France limits the assumptions and the candidates, we therefore indicated those that are most looking like our fossil wing. We do not want to be too affirmative on these points because of this difficulty.

4. Reference 2 and 3: Reference 3 should be Wang et al., the MS lost the rest co-authors; the sequence is incorrect, 2 and 3 should be switched each other; Yong-jie, not Yon-gjie.

Yes, thanks a lot, done

I would suggest to remove the reference 2. I do not think it is true mimicry. Those mecopterans from Daohugou displayed the normal pose with four wings slightly stretched out, as many mecopteran fossils in their preservation. It partly resembles ginkgo leaves but the major rejection due to their venation without any specialization. Moreover, those Daohugou mecopterans exhibit a constriction at wing base, an unusual character to be distinguished from ginkgo leaves.

We agree with this analysis, we have kept the reference, but added a short sentence.

5. Linguistic and format mistakes and other suggestions: 1) The authors may use more than 100 Ma instead of ca. 100 Ma, due to the age of Daohugou (with the previous earliest insect-plant mimicry fossil) no more than 165 Ma, and 265 Ma is just the boundary of Capitanian and Wordian;

Yes, thanks

2) demonstrates that the leaf-like homomorphous defense strategy was already... not necessarily "defense strategy", and it should be regarded as crypsis, or cryptic mimicry;

Yes, thanks

3) the Middle Permian should be change to Guadalupian.

Yes, done

4) SUPPLEMENTARY INFORMATION: The fossil was discovered by one of us (R.P.), should be P. R.;

Yes, done

5) in the red Permian of the Cians Formation... should be reworded

Yes, done

; 6) in table 1, the ratio of length/width etc. is "too accurate" that present 15 numbers after radix point.

Yes, changed, thanks

Reviewer #2 (Remarks to the Author):

The present manuscript deals with fascinating discovery of the earliest evidence of leaf-mimicking katydid from the Middle Permian deposits in France. This peculiar group of orthopteran insects was so far unknown from the Paleozoic strata and their oldest members come from Mesozoic. In addition the supportive morphometric analyses of fore wings (tegminae) based on extensive dataset of extant representatives of tettigoniids are given. PCA visualization of recent leaf-mimicking katydid subfamilies enabled Permotettigonia placing in the intersection of Pseudophyllinae and Pterophyllinae. Thus, the present contribution is original and novel in term of the earliest record of plant insect mimicry and parallel application of such complex landmark analysis of tegminae of recent katydid mimicking species.

1/ I recommend extending your review by indication of the taxa with supposed crypsis in other Paleozoic groups like particularly roachoids (Dictyoptera) with pteridosperm pinnules, other considered orthopterans (see Jarzembowski 1994, Prokop et al. 2015) and Triassic hemipterans (Cicadomorpha) presumably imitated buds, bracts, thorns, seeds, seed-bearing organs, or other small parts of their host plants (Shcherbakov 2011).

Yes, the plant mimicry by roachoids has been criticized by Jarzembowski.

The results of Nel et al are cited and discussed in Wedmann et al, which we cited in first. So we feel not necessary to add this reference. We have added Prokop et al's and Shcherbakov's one, thanks a lot.

Some of these putative members were put in doubt, but it should be briefly noticed in this contribution. Authors should note in their summary the earlier work dealing with an exceptional Cenozoic genus *Archepseudophylla* of leaf-mimicking katydid known from France (Nel et al. 2008).

This taxon is also listed in Wedmann et al, so it is not necessary to add the reference of Nel et al.

Perhaps it could be interesting to incorporate this well preserved Cenozoic genus in your landmark analysis and visualizes by PCA to see the position of another fossil genus.

Yes, this is a nice idea, we have done it

2/ I would prefer to cite several concrete examples with disruptive pattern of wing coloration like *Sinodunbaria* (Palaeodictyoptera) (e.g., Li et al. 2013) etc. instead of reference to the general textbook where colour patterns are omitted.

Yes, thanks, replaced

3/ I suggest keeping diagnosis in article as set of characters only and transfer the remaining comparison with other taxa into discussion as usual. Thus the main text will be clearly arranged.

Yes, done, thanks

To sum up, this work is an important contribution to our knowledge of plant insect ecology sparsely documented in Paleozoic. Besides a few above mentioned suggestions the manuscript is excellent. I fully support acceptance as publication in Nature Communication journal.

Please consider citing of the following papers:

Jarzembowski EA. 1994. Fossil cockroaches or pinnules insects? Proceedings of the Geologists' Association. 105:305–311.

Added

Li Y., Ren D., Pecharová M., & Prokop J. 2013. A new palaeodictyopterid (Insecta: Palaeodictyoptera: Spilapteridae) from the Upper Carboniferous of China supports a close relationship between insect faunas of Quilianshian (northern China) and Laurussia. *Alcheringa* 37(4):487-495.

Cited

Nel, A., Prokop, J. & Ross, A.J. 2008. New genus of leaf-mimicking katydids (Orthoptera: Tettigoniidae) from the Late Eocene - Early Oligocene of France and England. *C.R. Palevol*, 7 (4): 211-216.

Not necessary, see above

Prokop J., Szwedlo J., Lapeyrie J., Garrouste R. & Nel A 2015. New middle Permian insects

from Salagou Formation of the Lodève Basin in southern France (Insecta: Pterygota). *Annales de la Société Entomologique de France*, (N.S.) 51, 14-51.

Ok, added

Shcherbakov DE. 2011. New and little-known families of Hemiptera Cicadomorpha from the Triassic of Central Asia - early analogs of treehoppers and planthoppers. *Zootaxa* 2836:1–26.

yes, cited, thanks

REVIEWERS' COMMENTS:

Reviewer #1 (Remarks to the Author):

The major claim of the paper is concerning tracing back insect plant mimicry more than 100 million years than previously known. The paper would be interesting for palaeontologists, biologists, ecologists and the public at large. The conclusions of the paper are novel and convincing, and it would have great importance for plant-insect coevolution. I believe that the authors have made significant revision in the light of reviewers' suggestions. The paper in current form is suitable for publication in Nature Communications.

Reviewer #2 (Remarks to the Author):

I see my comments and suggestions were considered properly by the authors. Therefore, I have no additional requests.